# Effect of Vitamin D on Bone Regeneration: A Review

**DOI:** 10.3390/medicina58101337

**Published:** 2022-09-23

**Authors:** Giorgiana Corina Muresan, Mihaela Hedesiu, Ondine Lucaciu, Sanda Boca, Nausica Petrescu

**Affiliations:** 1Department of Oral Health, Iuliu Hatieganu University of Medicine and Pharmacy, 400012 Cluj-Napoca, Romania; 2Department of Oral Radiology, Iuliu Hatieganu University of Medicine and Pharmacy, 400006 Cluj-Napoca, Romania; 3Interdisciplinary Research Institute in Bio-Nano-Sciences, Babes-Bolyai University, 400271 Cluj-Napoca, Romania

**Keywords:** vitamin D, bone regeneration, osteoporosis, osseointegration, new bone formation

## Abstract

*Background and Objectives*: Vitamin D (Vit. D) is known for its role in the skeletal system. Vit. D deficiency is also widely researched for its effects on the healing of fractures, bone defects, and osseointegration of implants. In the literature, there are studies that investigated the effects of dietary supplementation with Vit. D to reduce Vit. D deficiency, but increasing the serum level of this vitamin takes time. Therefore, an attempt has been made to combat the effect of Vit. D deficiency through topical applications. The aim of this article was to conduct a review of the existing bibliographic data that investigate the effect of Vit. D on bone regeneration. *Materials and Methods*: In order to carry out this review, an electronic search was made in several databases and the articles found were selected and analyzed. *Results*: The in vitro studies’ results demonstrated that Vit. D has a high therapeutic potential by enhancing the differentiation of stem cells in osteoblasts. Human and animal studies were conducting using various methods, but most of them revealed that Vit. D has a positive influence on the process of bone regeneration. *Conclusions*: The overall results of the research showed that, indeed, Vit. D is beneficial for bone regeneration; however, most of the studies imply that a thorough research is still needed for finding the most effective mode of administration and the dose needed in order to achieve the desired effect.

## 1. Introduction

One of the challenges of maxillofacial and dentoalveolar surgery is bone regeneration. In the maxillofacial area, bone can be lost due to trauma, tooth loss, periodontal disease, and other pathologies [1]. 

Research in this area pays special attention to finding ways to regenerate lost bone mass.

Vitamin D (Vit. D) is a steroid hormone obtained mainly from sun exposure, diet, and dietary supplements. Vit. D is the name of two compounds, Vit. D2 (ergocalciferol) and Vit. D3 (cholecalciferol). Vit. D2 is produced by ultraviolet irradiation of ergosterol, and Vit. D3 results from ultraviolet irradiation of 7-dehydrocholesterol [2,3]. 

Because Vit. D is involved in bone metabolism and controls the immune system, implantology pays special attention to this vitamin. At the right concentration, the effect of this prohormone positively correlates with the process of osseointegration. Studies have shown that Vit. D has an important potential in the processes of regeneration of postoperative wounds, in dental implants osseointegration, and bone homeostasis around the implant [4]. 

Clinical studies have shown that Vit. D3 intake of 10 μg/day and calcium supplements (adjusted calcium intake of 1000 mg/day) not only reduce bone resorption and fracture rate, but also increase bone density and the total calcium in the body. However, studies exploring the distinct effect of systemic Vit. D3 supplementation and local application of Vit. D3 on bone tissue after traumatic fracture, pathological defects, or surgically created defects in large animals are rare [3,5]. 

The purpose of this review is to determine the possible link between the presence/level of Vit. D and bone regeneration.

## 2. Materials and Methods

In order to identify the relevant articles for this review, one author (G.C.M.) chose the search terms and performed an electronic search in the following databases: PubMed, Embase, Web of Science, and Google Scholar. The research included the articles published from 1963 until July 2022.

The search terms were as follows (Extended search terms are available in the Appendix A):–(bone regeneration) AND (Vit. D)–((Vit. D) AND (osseointegration)) AND (bone regeneration)

The screening of all the articles found in all the databases was performed by two independent authors. They screened the titles and abstracts of the articles and obtained full texts and performed further screening if the studies were deemed eligible. Disagreements were resolved by re-evaluating the articles or by consulting a third person.

Exclusion criteria: reviews and meta-analysis, articles not related to the topic, articles evaluating the effects of Vit. D on the articular or muscular system, and articles regarding the protein that binds Vit. D.

Inclusion criteria: articles evaluating the effect of Vit. D on bone regeneration, osseointegration, bone healing, studies on animals and humans, in vitro and in vivo studies, case–control clinical trials, and randomized controlled trials.

In order to assess the quality of the studies included in this review, the Newcastle–Ottawa scale (NOS) [6] was used, which assesses such quality characteristics of studies as selection, comparability and exposure; studies can receive a score between zero and nine stars (Table 1).

## 3. Results

The research preliminarily identified 1758 references to bone regeneration and the effects of Vit. D, of which 290 were found in PubMed, 325—in Web of Science, 965—in Embase, and 178—in Google Scholar.

After the exclusion of 900 duplicates/articles on another topic, the remaining 858 references were verified by examining the titles and abstracts in accordance with the inclusion and exclusion criteria described above. The remaining 58 articles were examined in full, and only 27 were eligible for our review (Figure 1).

Of the 27 articles included in the analysis, three studies were performed on patients, 17—on animals, and seven were in vitro studies.

Fifteen of the articles evaluated the regenerative effect of Vit. D, nine—osseointegration, three—the healing effect, and only one article reported that Vit. D does not have an effect on bone regeneration.

### 3.1. In Vitro Studies

The in vitro studies in this review aimed to evaluate the influence of Vit. D on fetal or adult osteoblasts, stem cells, whether administered *per se* or loaded onto nanoparticles and other nanostructures (Table 2).

In 2014, Rekha et al. [7] compared fetal and adult osteoblasts under the influence of Vit. D. The results of the study showed that fetal osteoblasts compared to adult osteoblasts undergo a significant increase in mineralization under the action of Vit. D, reflecting a great therapeutic potential in bone regeneration.

In 2018, Kim et al. [8] prepared osteoblast cells, to which they added 1.25-dihydroxyvitamin D3 in order to evaluate the effect of this active form of Vit. D on osteoblasts. The research results showed that 1.25-dihydroxyvitamin D3 had a positive effect on these cells; more precisely, it could function as a stimulating factor in bone regeneration.

In 2020, Abdelgawad et al. and Petrescu et al. [12,13] showed the positive effects of Vit. D alone or in combination with other substances or under the action of a laser on stem cells (dental or from the periodontal ligament), stimulating their differentiation to odontoblasts.

In 2019–2020, Nah et al., Chen et al., and Mahdavi et al. [9,10,11] conjugated Vit. D onto nanoparticles and other functional nanocarriers to enhance its bioactivity. The results of the studies showed the potential of these nanomaterials to improve osteogenic differentiation and bone regeneration.

### 3.2. Studies Carried out on Animals

In our review, we included 17 studies on animal subjects, specifically on rats, mice, and dogs (Table 3).

In 1980, Galus et al. [14] demonstrated that Vit. D derivatives 24R,25-dihydroxyvitamin D and 1 alpha-hydroxyvitamin D accelerate the rate of appositional growth, increase the percentage of existing osteoid connections, and decrease the number, the width, and the perimeter of new osteoid connections. The authors also reported that 1 alpha-hydroxyvitamin D results in multidirectional bone remodelling that increases mineralization in concert with resorption while 24R,25-dihydroxyvitamin D stimulates bone formation and mineralization, reducing bone resorption.

In 2009, Kelly et al. [15] demonstrated that Vit. D deficiency affects the osseointegration of implants. 

In the same year, Uysal et al. [16] showed the beneficial effect of a Vit. D analog in bone regeneration. They expanded the palatal median suture and applied ED-71 locally in a single dose of 0.8 μg/kg body weight.

In 2012, Hong et al. [17] showed through research on dogs that the post-extraction use of a combination of biphasic phosphate alloplastic material applied topically and Vit. D and calcium administered orally resulted in positive effects in terms of bone regeneration.

Dvorak et al. and Zhou et al. [18,19] published in 2012 the results of a study performed on ovariectomized rats. Their aim was to demonstrate the negative effects of Vit. D deficiency and the beneficial effect of Vit. D supplementation on the osseointegration of implants.

In 2014, Liu W et al. [20] investigated the effect of Vit. D on implant fixation in subjects with chronic kidney disease. The authors showed that by injecting Vit. D 8 weeks after the second intervention, they obtained an improved bone-to-implant ratio and an increased bone volume around the implant.

In 2015, Fügl et al. [22] showed in their results that Vit. D deficiency would not affect bone regeneration, and the administration of a single dose of calcitriol did not improve bone regeneration. A few months later, in the same year, Liu H et al. [21] used calcitriol-soaked collagen membranes which they applied over bone defects. The results of their research showed that these membranes accelerated bone formation and maturation.

In the same year, Hong et al. [23] supported the positive effect of Vit. D on bone regeneration. They evaluated the effects of the combination of topical application of calcitriol and oral administration of Vit. D on regeneration and bone density. The results showed that topical application of calcitriol accelerated the formation of new bone, increased bone density, but, compared to systemic Vit. D administration, topically applied calcitriol had a lower effect. 

Another study that evaluated the effects of Vit. D was that of Salomó-Coll et al. [24,27] from 2016 along with that of 2018 by the same authors. They showed that soaking the implant surface with Vit. D before intraosseous application reduced the loss of crestal bone and increased the contact of the implant with the bone by 10%. 

In 2017, Fischer et al. [25] evaluated whether Vit. D deficiency compromised bone regeneration and whether early Vit. D administration had beneficial effects on healing. Their results suggested that Ca/Vit. D supplementation initiated at the time of fracture offset the negative effects of Vit. D deficiency.

Another study conducted in 2017 by Han et al. [26] analyzed histological changes after systemic administration of eldecalcitol (ELD) to rats in correlation with bone regeneration. Their research found that taking ELD improved the growth of new bone mass.

In 2019, Wang et al. [29] designed two new Vit. D derivatives, 25-dihydroxyvitamin D3 and α,α-difluoro-cyclohexanone, which, according to the research, greatly improved the parameters related to bone mass and density.

In 2020, Cignachi et al. [30] assessed whether the administration of Vit. D, insulin, or Vit. D and insulin influenced bone regeneration in mice with type 1 diabetes. Their research showed that all mice regardless of sex had a similar detriment in terms of bone regeneration, and the administration of Vit. D or insulin improved bone healing.

### 3.3. Clinical Studies

The clinical trials included in the review consisted of three prospective randomized controlled trials, one of which was double-blind, and the other two had convenience sampling (Table 4).

In 1989, Zarubina et al. [31] investigated the efficacy and differences in the action of some Vit. D derivatives (1-alpha-hydroxycholecalciferol and 1-alpha,25-dihydroxycholecalciferol) in bone regeneration in patients with osteoporosis/osteomalacia. The results demonstrated the positive effects of the two derivatives in terms of regeneration, their effectiveness being equal. 

In 2019, Grønborg et al. [32] assessed the effect of eating Vit. D-fortified foods in a clinical study. That study showed that Vit. D-fortified foods did not improve bone markers, and muscle strength did not change significantly. 

In 2021, Kwiatek et al. [33] presented the results of their study which evaluated the effect of 25-hydroxycholecalciferol along with the treatment of Vit. D deficiency on changes in bone levels around the implant. The study found that a correct dose on the day of surgery and adequate treatment of Vit. D deficiency had a positive effect on the increase in bone levels around the implant during osseointegration.

## 4. Discussion

The studies on animals showed positive effects of the administration of Vit. D, either systemically or locally. Four of these studies showed the impact of Vit. D on implants osseointegration. Salomó-Coll et al. [24,27] showed that by soaking the implants in 10% Vit. D solution before their intraosseous application, the rate of increase of the contact surface between the implant and the bone could be more than 10%. Kelly et al. and Dvorak et al. [15,18] showed that Vit. D deficiency has a negative influence on the osseointegration of implants, an effect that can be reversed by the administration of Vit. D. Zhou et al. [19] demonstrated that by administering 0.1 μg/kg/day 1.25(OH)2D3 by oral gavage for 8 weeks in rats with induced osteoporosis, they obtained an improvement in the osseointegration of the implants. The rest of the animal studies highlighted the positive effect that Vit. D has on bone regeneration, either by oral administration or by local application [5,15,21,23,25]. One single study by Fügl et al. [22] stated that Vit. D deficiency does not necessarily affect bone regeneration and that a single topical application of calcitriol does not improve healing. The study conducted by Cignachi et al. [30] showed that Vit. D improves bone regeneration even in rats with induced diabetes.

The studies on patients represented by three clinical trials evaluated the effects of Vit. D. Zarubina et al. [31] showed that Vit. D derivatives 1-alpha-hydroxycholecalciferol and 1-alpha,25-dihydroxycholecalciferol have positive effects on bone regeneration, with no differences between them. The study by Kwiatek et al. [33] reported that Vit. D administration before implant surgery associated with the treatment of Vit. D deficiency increased bone levels in implants, improving osseointegration. On the other hand, Grønborg et al. [32] showed that the consumption of foods fortified with Vit. D for twelve weeks did not bring significant changes in terms of markers of bone turnover and muscle strength.

The in vitro studies provided further evidence of the positive effects of Vit. D on osteoblasts and stem cell differentiation. These studies showed that osteoblasts together with Vit. D have a great potential in bone regeneration and, in the case of stem cells, the presence of Vit. D improved the differentiation to osteoblasts [11]. All the in vitro studies brought us useful information on the use of nanostructures/nanoparticles loaded with Vit. D acting as reservoirs of the active substance, thus improving bone regeneration [9,10].

The main limitation of this review was the small number of articles available in the literature and their heterogeneity—clinical studies, animal model or human patient studies, in vitro studies, different methods of investigation, different design, etc.

Heterogeneity and different study designs prevented us from making a clear statement about the influence of Vit. D on bone regeneration because its effects might be influenced by the presence or deficiency of other vitamins or elements (vitamin K). 

Epidemiological studies suggest that the absence of vitamin K in the organism is correlated with bone disease and mineralization issues. [34] It is known that vitamin K is a catalyst for the process of carboxylation of certain proteins. Osteocalcin and matrix Gla protein are two calcium-binding vitamin K-dependent proteins [35]. The necessity of these proteins for bone health has been proven in previous studies [36], but there is also evidence that bone mineralization is not influenced by the absence of osteocalcin [37]. Thus, future studies need to be conducted, particularly on organisms deficient in several elements considered important for bone formation, in order to identify the exact elements required in the process.

In vitro studies also provide valuable information about the effects of Vit. D, but they are carried out in a controlled environment, without other interferences that would be found in the human body (such as vitamin deficiency). Since bone health depends on various vitamins and nutrients (calcium, magnesium, phosphorus, fluorides, copper, zinc, boron, manganese, potassium, vitamin A, vitamin C, B vitamins) [38], additional studies are needed with and without these elements besides Vit. D that have been proven in other studies to influence the bone regeneration process.

As shown above, to assess the risk of bias, the NOS was used to assess the items as follows: low risk of bias (7–9 NOS score), high risk of bias (4–6 NOS score), and very high risk of bias (0–3 NOS score), the vast majority of the articles included in this review have a score of 7,8—low risk of bias, and only seven had high risk of bias.

## 5. Conclusions

According to the evidence in the studies included in this review, we concluded that Vit. D has a major role in bone regeneration, for proper bone mineralization. In the context in which today’s society has a growing need for bone reconstruction and dental implants, for the modern patients who want fast results, Vit. D administration could increase the rate of osseointegration using a minimally invasive and inexpensive technique.

## Figures and Tables

**Figure 1 medicina-58-01337-f001:**
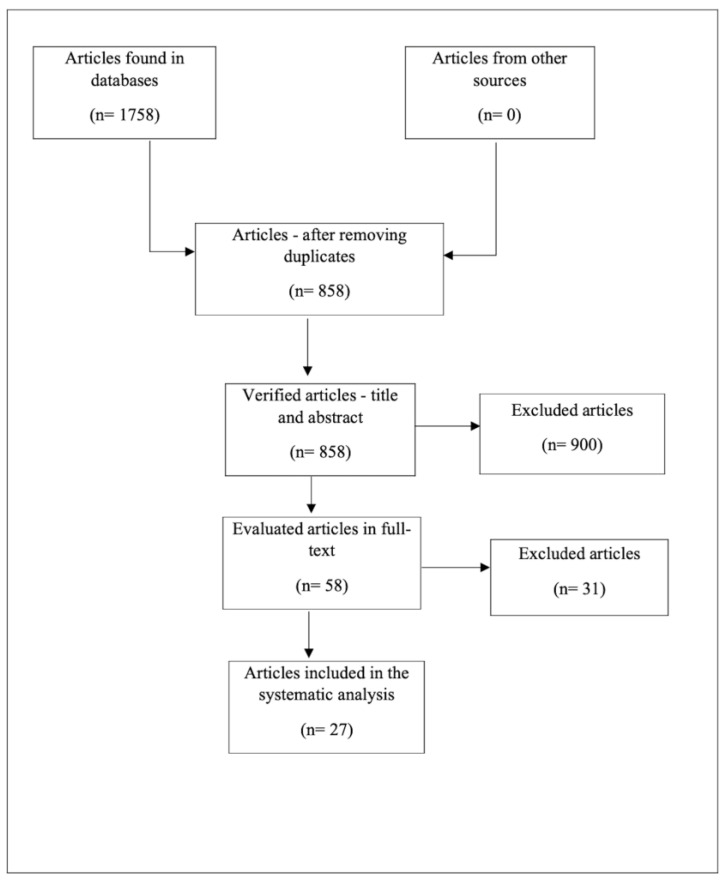
Diagram representing the inclusion of studies.

**Table 1 medicina-58-01337-t001:** The Newcastle–Ottawa scale.

Study	Selection	Comparability	Exposure	Score
Galus et al.	***	**	***	8
Uysal et al.	***	*	***	7
Hong et al.	***	*	**	6
Dvorak et al.	***	*	**	6
Zhou et al.	***	*	***	7
Liu et al.	***	*	***	7
Liu et al.	***	*	***	7
Fügl et al.	***	*	**	6
Hong et al.	***	*	**	6
Salomó-Coll et al.	***	*	***	7
Fischer et al.	***	*	***	7
Han et al.	***	*	***	7
Salomó-Coll et al.	***	**	***	8
Won et al.	***	*	***	7
Cignachi et al.	***	**	***	8
Zarubina et al.	0	**	**	4
Grønborg et al.	***	*	**	6
Kwiatek et al.	***	*	**	6
Kim et al.	***	*	***	7
Wang et al.	***	**	***	8
Nah et al.	***	**	***	8
Chen et al.	0	*	***	4
Mahdavi et al.	0	*	***	4
Petrescu et al.	0	*	***	4
Abdelgawad et al.	***	**	***	8

For each parameter, there is a maximum number of stars which can be given using the questionnaire in the Newcastle–Ottawa scale manual: *selection* (****), *comparability* (**), and *exposure* (***). The number of stars for each of the previously mentioned categories, were given considering the criteria in the Newcastle-Ottawa scale. For *selection*, the maximum number of stars is 4 (****), but none of the studies included in this review have accumulated the maximum number of stars for that category. Thus, in the first column, the maximum number of stars given is 3 (***). For *comparability*, the maximum number of stars is 2 (**), some of the studies met the criteria to get 2 stars (**), others did not, getting only one star (*). For *exposure*, the maximum number of stars which can be given for each study is 3 (***). Some of the studies met the criteria and got 3 stars (***) and some only deserved 2 stars (**). If a study did not meet the criteria for a parameter, it received 0 stars. The total number of stars of each article is in the “Score” column could be from 0 to 9. A high number of stars represents a high quality of the study.

**Table 2 medicina-58-01337-t002:** Characteristics of the in vitro studies and their results.

Author, Year [Reference]	Main Objective	Intervention, Dose, and Frequency	Results
Rekha et al, 2014 [7]	Evaluation of the in vitro efficacy of Vit. D on osteoblastic activity in both fetal and adult osteoblasts	1. Collection of samples: – fetal osteoblasts—calvariae and long bones from the fetuses of two pregnant rabbits – adult osteoblasts—radii from two adult white rabbits 2. Isolation, culture, extension, and characterization 3. Placement of viable cells in the osteogenic environment with or without 1.25-dihydroxyvitamin D3	Fetal osteoblasts compared to adult osteoblasts have shown a significant increase in mineralization upon the addition of Vit. D. This reflects a high therapeutic potential of fetal osteoblasts along with Vit. D3 in bone regeneration.
Kim et al, 2018 [8]	Evaluation of the effects of 1.3-dihydroxyvitamin D3 on the proliferation, differentiation, and mineralization of the matrix of osteoblast-like MC3T3-E1 cells in vitro	MC3T3-E1 osteoblastic cells and 1.25-dihydroxyvitamin D3 were prepared	The authors suggest that 1.25-dihydroxyvitamin D3 positively affects cell differentiation and matrix mineralization. Therefore, it can function as a stimulating factor in the formation of osteoblastic bone and can be used as an additive in the treatment of bone regeneration.
Nah et al, 2019 [9]	Synthesis of conjugated GNPs (gold nanoparticles) with VGNPs (vitamin D-conjugated GNPs) to allow improved osteogenesis	Synthesizing GNPs conjugated with VGNPs	VGNPs can be applied as potent carriers that enhance osteogenic differentiation. The results of this study could help design a nanoparticle system for the treatment of osteoporosis in the field of bone tissue engineering.
Chen et al, 2020 [10]	Construction of a biofunctional multilayer structure containing Vit. D and calcitonin (CT) on a titanium alloy implant (Ti6Al7Nb)	1. Molecules of β-cyclodextrin (β-CD) molecular reservoirs grafted on chitosan molecules and loaded with calcitriol (Vit. D) 2. Molecular complex co-assembled with calcitonin (CT) 3. Ti6Al7Nb substrate	In vitro results show that the released Vit. D and CT individually regulated the expression of the calcium-binding protein (including calbindin-D9k and calbindin-D28k) and BMP2 in osteoblasts in peri-implant regions to stimulate their deposition and differentiation from Ca. Micro-CT results and in vivo histological analyses also demonstrate that a coloaded Vit. D/CT implant can dramatically improve bone remodelling under osteoporosis.
Mahdavi et al, 2020 [11]	The aim of the study was to obtain new scaffolds with drug release capability usable in bone tissue engineering	Manufacturing of graphene (GO) oxide scaffolds loaded with gelatin (G)–hydroxyapatite (HA)–Vit. D with different concentrations using the solvent casting method	The results demonstrated the potential of these scaffolds to induce bone regeneration.
Petrescu et al, 2020 [12]	To establish a new differentiation protocol using cannabidiol (CBD) and Vit. D for better and faster osteogenic differentiation of mesenchymal stem cells (MSCs) derived from dental tissue	1. MSC harvesting, isolation, and characterization 2. Evaluation of the effects of CBD and Vit. D in terms of osteogenic differentiation of stem cells	This study provides evidence for a better understanding of the effects of CBD and Vit. D on MSC populations of dental origin, supporting the development of tissue engineering in the field of dentistry.
Abdelgawad et al, 2020 [13]	Evaluation of the effects of photobiomodulation and Vit. D (as an anabolic factor) on HPDLSCs (human periodontal ligament stem cells) for bone regeneration	1. Collection, isolation, and characterization of periodontal ligament stem cells 2. Their division into six groups: groups I and II, control and (10^−7^ Mol) vitamin D, respectively; group III, irradiation at 1 J/cm^2^; group IV, irradiation at 1 J/cm^2^ and culture with Vit. D; group V, irradiation at 2 J/cm^2^; group VI, irradiation at 2 J/cm^2^ and culture with Vit. D	Laser irradiation at 2 J/cm^2^ combined with Vit. D improved osteoblast differentiation and proliferation of the cultured HPDLSCs.

**Table 3 medicina-58-01337-t003:** Characteristics of the studies on animals and their results.

Author, Year [Reference]	Main Objective	Number/Type of Animals	Intervention, Dose, and Frequency
Galus et al, 1980 [14]	To assess whether the two forms of Vit. D have an effect on bone remodelling, whether there is a difference between them, and how the difference is expressed	8 mongrel dogs	1. Vit. D deprivation, bone excision, study on dogs 2. Administration of 24R,25-dihydroxyvitamin D orally in doses of 0.09–0.14 btg/kg body weight, 1-alpha-hydroxyvitamin D—0.08–0.12 btg/kg body weight every 2 days for 15 weeks.
Kelly et al, 2009 [15]	Assessing the effect of Vit. D deficiency on implant osseointegration	Male Sprague–Dawley rats	1. Two groups of rats, control group and experimental group 2. Experimental group—Vit. D deprivation 3. Application of treated mini-implants into the femur
Uysal et al, 2009 [16]	Evaluation of the effects of ED-71, an active analog of Vit. D, on bone regeneration	16 Wistar rats	1. Expansion of the mid-palatal suture in rats 2. Vehicle solution administration to the control group 3. Experimental group—single dose of ED-71, 0.8 μg/kg body weight, topical application
Hong et al, 2012 [17]	Evaluation of the potential effects of the combination of topical application of biphasic calcium phosphate alloplastic material and oral administration of Vit. D and calcium	9 beagle dogs	1. Extraction of four mandibular premolars in dogs 2. Making of four dental sockets of which only two grafts with biphasic calcium phosphate alloplastic material 3. Random distribution of the subjects into two groups: case and control 4. Administration of Vit. D/calcium to the experimental group
Dvorak et al, 2012 [18]	Evaluation of the effects of systemic Vit. D supplementation on implant osseointegration	50 Sprague–Dawley rats	1. Three groups of ovariectomized rats 2. Experimental group—diet without Vit. D 3. Application of two mini-implants in the tibia
Zhou et al, 2012 [19]	Investigation of the effect of 1.25(OH)2D3 on osseointegration in osteoporotic rats	20 Sprague–Dawley rats	1. Bilateral ovariectomy in rats 2. Application of two screws to the proximal tibia 3. Randomized distribution of the subjects in two groups 4. Administration of 1.25 (OH)2D3 orally at 0.1 μg/kg/day
Liu W et al, 2014 [20]	Evaluation of the effect of Vit. D on implants osseointegration in CKD (chronic kidney disease) mice.	30 C57BL mice	1. Induction of CKD by nephrectomy 2. Creation of three groups of subjects: control, CKD, and CKD + Vit. D 3. In the CKD + Vit. D injection group, at 8 weeks from the second intervention, Vit. D-100 ng/kg body weight three times a week until slaughter 4. Application of implants
Liu H et al, 2015 [21]	Investigating the influence of calcitriol on osseoinduction after local administration in mandibular bone defects	96 Wistar rats	1. Loading collagen membranes with calcitriol 2. Making up the control and experimental groups 3. Applying membranes in the defects
Fügl et al, 2015 [22]	Evaluation of the effect of Vit. D deficiency and local calcitriol application on bone regeneration	60 Sprague–Dawley rats	1. Distributing subjects in three groups—two experimental groups and one control group 2. Making bone defects 3. Application of calcitriol-soaked collagen in one of the experimental groups
Hong et al, 2015 [23]	Exploring the regenerative potential of calcitriol in local applications and comparing possible regeneration effects in association with systemic Vit. D administration	10 beagle dogs	1. Random division of dogs into two groups, non-Vit. D/C and Vit. D/Ca, receiving Bio-Cal supplements 2. Preparation of sockets 3. Grafting of experimental sockets with 1 mL Calcijex^®^ and 0.5 g HA (hydroxyapatite)/β-TCP
Salomó-Coll et al, 2016 [24]	Evaluation of the effect of Vit. D when applied to the implant surface	6 American foxhound dogs	1. Carrying out the extractions 2. The test group consisted of 12 implants soaked in 10% Vit. D solution
Fischer et al, 2017 [25]	Assessing whether vitamin D deficiency compromises bone spotting and leads to its loss, as well as whether Vit. D administration immediately after fracture increases healing	24 C57BL/6 J mice	1. Female ovariectomized mice 2. Three groups: one control group and two experimental groups (diet without vitamin D/calcium) 3. Femoral osteotomy 4. The experimental group immediately received a diet supplemented with Vit. D/Ca S8276-E712, 2.0% calcium, 2000 IU/kg Vit. D, Sniff
Han et al, 2017 [26]	Observing histological changes after systemic administration of eldecalcitol (ELD) in combination with guided bone regeneration during the healing of bone defects in rats	64 Wister rats	1. Two groups of rats 2. Creating bone defects in the femur in all the rats 3. Eldecalcitol (50 ng/kg body weight) administered to the experimental group
Salomó-Coll et al, 2018 [27]	Evaluation of osseointegration of implants in case of topical applications of melatonin, Vit. D.	6 American foxhound dogs	1. Extractions of the distal roots of the lower premolars 2. Application of three bilateral implants 3. Formation of three groups: control group implants (CI), group MI—implants soaked in melatonin, group DI—implants soaked in Vit. D
Won et al, 2019 [28]	Evaluation of the effect of shiitake mushroom fortified with Vit. D on both 25(OH)D and calcium serum levels	48 Sprague–Dawley rats	1. Simultaneously operated rats: simulated (sham) and ovariectomized (OVX) divided into three groups 2. Control group—sham and Vit. D-deficient diet – UV (X)—OVX group under a nonirradiated mushroom powder diet – UV (O)—OVX group under an irradiated mushroom powder diet
Wang et al, 2019 [29]	Design, synthesis, and evaluation of two novel products 1α,25-dihydroxyvitamin D3 containing a portion of α,α-difluoro-cyclopentanone (3) or α,α-difluoro-cyclohexanone (4)	40 Wistar rats	1. Synthesis of compounds 3 and 4 2. Biological studies: five groups: – sham group – OVX group – positive drug group – 200 ng/kg/day CAL for 6 weeks – 25 ng/kg/day compound 3 or 4 by gavage for 6 weeks
Cignachi et al, 2020 [30]	Assessment of bone regeneration of a femoral defect in mice with type 1 diabetes according to sex, presence of insulin, insulin associated with Vit. D	186 C57BL/6J mice	1. Induction of type 1 diabetes in mice: five daily injections of streptozotocin (STZ; 50 mg/kg) 2. Creating femoral defects 3. Treatment: Male and female mice in the control group or T1D were randomly assigned to four subgroups depending on treatment: (i) vehicle; (ii) Vit. D; (iii) insulin; (iv) Vit. D plus insulin. Dosage: Vit. D—4 μg/kg orally; insulin—3 IU/kg subcutaneously

**Table 4 medicina-58-01337-t004:** Characteristics of the studies on patients and their results.

Author, Year [Reference]	Main Objective	Number of Patients	Intervention, Dose, and Frequency	Results
Zarubina et al, 1989 [31]	Evaluating the effectiveness of these Vit. D derivatives in regenerating bone structures in patients with osteoporosis/osteomalacia	–	1. Patients with osteoporosis/osteomalacia 2. Administration of 1-alpha-hydroxycholecalciferol and 1-alpha,25-dihydroxycholecalciferol at 0.25–2 μg daily	The evaluated derivatives had positive effects in terms of regeneration, their effectiveness being equal.
Grønborg et al, 2019 [32]	Investigating the effect of 12 weeks of intervention with Vit. D-fortified foods on markers of bone turnover and muscle strength	143 women	1. Women of Pakistani and Danish origin 2. Distribution of women in four groups by randomization 3. Administration of placebo or fortified foods with 30 μg Vit. D/day for 12 weeks	Consumption of food fortified with Vit. D for 12 weeks did not lead to significant changes in bone turnover markers, and muscle strength did not change significantly after the intervention.
Kwiatek et al, 2021 [33]	Assessing the effect of the 25-hydroxycholecalciferol concentration and treatment of Vit. D deficiency on changes in bone at the implant site during the process of osseointegration in the mandible	122 patients	1. Patients with edentation in the premolar–molar area divided into three groups 2. Group A treated with <30 ng/mL of 25-hydroxycholecalciferol in blood serum without Vit. D supplementation; Group B treated with <30 ng/mL of 25-hydroxycholecalciferol in blood serum, with Vit. D supplementation; Group C treated with normal Vit. D level in blood serum and placebo	The correct level of 25-hydroxycholecalciferol on the day of surgery and treatment of Vit. D deficiency had a significant influence on the increase in bone level at the implant site during the osseointegration process.

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
