# Peer review of "Effect of Vitamin D on Bone Regeneration: A Review"

_medicina, 2022, doi:10.3390/medicina58101337_

Round 1
Reviewer 1 Report
I agree most of the discussion in this review article.
My comment is as below:
In the literature, just as mentioned by the author, some said that Vitamin D deficiency would not affect bone regeneration and the administration of calcitriol did not improve bone regeneration. However, some said the opposite.
Although vitamin D enhance the differentiation of the stem cells in osteoblasts which produce osteocalcin. However, the fact is that only carboxylated osteocalcin can carry calcium into the bone. Carboxylation of the osteocalcin is vitamin K2 dependent. If the subject is vitamin K2 deficient, then vitamin D might have no function on bone regeneration.
Suggest the authors make more discussion in this point of view.
Author Response
Reviewer 1
( ) I would not like to sign my review report
(x) I would like to sign my review report
English language and style
( ) Extensive editing of English language and style required
( ) Moderate English changes required
( ) English language and style are fine/minor spell check required
(x) I don't feel qualified to judge about the English language and style
Comments and Suggestions for Authors
I agree most of the discussion in this review article.
My comment is as below:
In the literature, just as mentioned by the author, some said that Vitamin D deficiency would not affect bone regeneration and the administration of calcitriol did not improve bone regeneration. However, some said the opposite.
Although vitamin D enhance the differentiation of the stem cells in osteoblasts which produce osteocalcin. However, the fact is that only carboxylated osteocalcin can carry calcium into the bone. Carboxylation of the osteocalcin is vitamin K2 dependent. If the subject is vitamin K2 deficient, then vitamin D might have no function on bone regeneration.
Suggest the authors make more discussion in this point of view.
Thank you very much for the time you dedicated to this manuscript and for appreciating our work.
Heterogeneity and different study designs prevent us from making a clear statement about the influence of Vit. D on bone regeneration, because its effects might be influenced by the presence or deficiency of other vitamins or elements (vitamin K).
Epidemiological studies suggest that the absence of vitamin K in the organism is correlated with bone disease and mineralization issues. [34] It is known that vitamin K is a catalyst for the process of carboxylation of certain proteins. Osteocalcin and matrix Gla protein are two calcium-binding vitamin K-dependent proteins [35]. The necessity of these proteins for bone health has been proven in previous studies [36] but there is also evidence that bone mineralization is not influenced by the absence of osteocalcin [37]. Thus, future studies need to be done, particularly on organisms deficient in several elements considered important for bone formation, in order to identify the exact elements required in the process.
In vitro studies also provide valuable information about the effects of Vit. D, but they are carried out in a controlled environment, without other interferences that would be found in the human body (like vitamin deficiency). Since bone health depends on various vitamins and nutrients (Calcium, Magnesium, Phosphorus, Flouride, Copper, Zinc, Boron, Manganese, Potassium, Vitamin A, Vitamin C, B Vitamins) [38], additional studies are needed, with and without these elements, besides Vit. D, that have been proven in other studies to influence the bone regeneration process.

Reviewer 2 Report
In my opinion, the work is written well. There are minor editorial errors and one issue that needs to be clarified. Please see my comments.
L54 ‘’GCD’’ - Can't find an author to whom the initials would match. Please check.
L56 – ‘’ (2022).’’ - please add the exact month of completion of the review.
L58-72 – ‘’ Search terms:…” In my opinion, the work will be clearer as these data will be in the supplementary materials.
Please explain in place L54 you write that the analysis was made by one author and in place L73 that two. Please describe more clearly.
Why was the review not registered in the PROSPERO database?
L86 - I suggest that the title be written without the abbreviation.
L86 - Under the table, please explain the designations used in it:0 and *, **, ***.
L240 -241 - This sentence in the conclusion sounds like a repetition of the results. Please rephrase.
Author Response
Reviewer 2
(x) I would not like to sign my review report
( ) I would like to sign my review report
English language and style
( ) Extensive editing of English language and style required
( ) Moderate English changes required
(x) English language and style are fine/minor spell check required
( ) I don't feel qualified to judge about the English language and style
Comments and Suggestions for Authors
In my opinion, the work is written well. There are minor editorial errors and one issue that needs to be clarified. Please see my comments.
L54 ‘’GCD’’ - Can't find an author to whom the initials would match. Please check.
Thank you for your constructive feedback. There was also a redaction mistake in the text and we apologise for that. It should have been GCM instead of GCD. We’ve changed it in the manuscript as follows:
„In order to identify the relevant articles for this review, one author (GCM) chose the search terms and did the electronic search on the following databases: PubMed, Embase, Web of Science and Google Scholar.”
L56 – ‘’ (2022).’’ - please add the exact month of completion of the review.
Thank you for your suggestion, we included the month.
„The research included articles published from 1963 up to present (July, 2022).”
L58-72 – ‘’ Search terms:…” In my opinion, the work will be clearer as these data will be in the supplementary materials.
Thank you for your suggestion. As recommanded by the reviewer, we replaced the extended version of the search terms in the manuscript with a shorter version. The extended version of the search terms is in a different document, as a supplementary file, which we’ll send by email to the Assistant Editor.
„The searching strategy was the following:
Search terms: - (bone regeneration) AND (Vit. D)
- (Vit. D) AND (osseointegration) AND (bone regeneration).”
Please explain in place L54 you write that the analysis was made by one author and in place L73 that two. Please describe more clearly.
Thank you for stating that the afirmations were not clear enough. We rephrased it:
„In order to identify the relevant articles for this review, one author (GCM) chose the search terms and did the electronic search on the following databases: PubMed, Embase, Web of Science and Google Scholar.”
„The screening of all the articles found on all databases was done by two independent authors. They screened the titles and abstracts of the articles and they obtained full texts and performed further screening when studies were deemed eligible.”
Why was the review not registered in the PROSPERO database?
Thank you for your question, our article is a simple review, not a systematic review. According to our knowledge, only the systematic reviews need to be registered in the PROSPERO database.
L86 - I suggest that the title be written without the abbreviation.
Thank you for your suggestion, as recommended, we wrote the name of the table without the abbreviation.
„Table 1. The Newcastle-Ottawa Scale”
L86 - Under the table, please explain the designations used in it: 0 and *, **, ***.
We explained under the table the meaning of the designations of 0, *, **, ***, as follows:
„For each parameter, there is a maximum number of stars which can be given, using the questionaire in the Newcastle-Ottawa Scale manual; Selection (****), Comparability (**) and Exposure (****). If a study did not meet the criteria for a parameter, it received 0 stars. The total number of stars of each article is in the „Score” column and could be from 0 to 8. A high number of stars represents a high quality of the study.”
L240 -241 - This sentence in the conclusion sounds like a repetition of the results. Please rephrase.
Thank you for your suggestion. We rephrased as below:
„According to the evidence in the studies included in this review, we conclude that Vit. D has a major role in bone regeneration, for a proper bone mineralization.”
